# ADSC-Based Cell Therapies for Musculoskeletal Disorders: A Review of Recent Clinical Trials

**DOI:** 10.3390/ijms221910586

**Published:** 2021-09-30

**Authors:** Seahyoung Lee, Dong-Sik Chae, Byeong-Wook Song, Soyeon Lim, Sang Woo Kim, Il-Kwon Kim, Ki-Chul Hwang

**Affiliations:** 1Institute for Bio-Medical Convergence, College of Medicine, Catholic Kwandong University, Gangneung 210-701, Korea; sam1017@ish.ac.kr (S.L.); songbw@cku.ac.kr (B.-W.S.); slim724@cku.ac.kr (S.L.); swk74@cku.ac.kr (S.W.K.); 2Department of Orthopedic Surgery, International St. Mary’s Hospital, Catholic Kwandong University, Gangneung 210-701, Korea; drchaeos@gmail.com

**Keywords:** adipose-derived stem cell, clinical trials, musculoskeletal disorders

## Abstract

Recently published clinical trials involving the use of adipose-derived stem cells (ADSCs) indicated that approximately one-third of the studies were conducted on musculoskeletal disorders (MSD). MSD refers to a wide range of degenerative conditions of joints, bones, and muscles, and these conditions are the most common causes of chronic disability worldwide, being a major burden to the society. Conventional treatment modalities for MSD are not sufficient to correct the underlying structural abnormalities. Hence, ADSC-based cell therapies are being tested as a form of alternative, yet more effective, therapies in the management of MSDs. Therefore, in this review, MSDs subjected to the ADSC-based therapy were further categorized as arthritis, craniomaxillofacial defects, tendon/ligament related disorders, and spine disorders, and their brief characterization as well as the corresponding conventional therapeutic approaches with possible mechanisms with which ADSCs produce regenerative effects in disease-specific microenvironments were discussed to provide an overview of under which circumstances and on what bases the ADSC-based cell therapy was implemented. Providing an overview of the current status of ADSC-based cell therapy on MSDs can help to develop better and optimized strategies of ADSC-based therapeutics for MSDs as well as help to find novel clinical applications of ADSCs in the near future.

## 1. Introduction

Stem cells refer to a group of unspecialized cells with the ability to differentiate into many lineage-specific cell types and to renew themselves. Although embryonic stem cells are known to have the most powerful pluripotency [1], their ethical issues and limited availability have promoted the search for adult stem cells for tissue regeneration and stem-cell-based therapeutics [2]. One of the well-known examples of such adult stem cells are bone-marrow-derived mesenchymal stem cells (BM-MSCs), and since their first discovery in 1970 [3], they have been considered the major players in stem-cell-based therapies, being the most frequently used cells in clinical settings [4]. However, the invasive harvesting procedure of BM-MSC poses unnecessary pain and/or risk of infection, and it may also yield an insufficient amount of cells for clinical applications [5]. Such drawbacks of BM-MSCs have driven yet another search, and a number of adult stem cells from different sources, such as adipose tissue, umbilical cord, dental pulp, and endometrium, have been reported [6]. Among these cells, adipose-derived stem cells (ADSCs) are considered good candidates for autologous cell therapy since they can be obtained in high numbers from the abundant adipose tissue of the body [7].

Since the very first isolation and identification of human ADSCs in 2002 [8], numerous strategies to utilize ADSCs as a main component of regenerative cell therapeutics have been developed and tested. As the name indicates, ADSCs refer to adult mesenchymal stem cells obtained from adipose tissue. In terms of their characteristics, very similar to the BM-MSCs, they possess a self-renewal ability and multi-potency. On the other hand, unlike the BM-MSCs, a sufficient amount of ADSCs can be easily obtained from adipose tissue with a minimally invasive procedure such as liposuction, and adherent ADSCs can be expanded in vitro, maintaining the capacity to differentiate [9]. Such ease of harvesting and multi-potency of ADSCs make them attractive adult stem cells for repairing damaged tissues and organs, and the PubMed search for recently published clinical trials (within the last 10 years) involving the use of ADSCs indicated that approximately one-third of the published clinical studies were conducted on musculoskeletal disorders (MSD). 

MSD refers to a wide range of degenerative conditions of joints, bones, and muscles. The most common examples of MSD include osteoarthritis, osteoporosis, rheumatoid arthritis, and sports injuries, and these conditions are also the most common causes of chronic disability worldwide, being a major burden to society [10]. Conventional treatment modalities for MSD such as pharmacological and non-pharmacological therapies are used mainly to reduce the pain associated with these conditions. In other words, these treatment options may relieve the symptoms and the pain associated with musculoskeletal disorders, but they are often associated with a wide range of undesirable side effects and are not sufficient to correct the underlying structural abnormalities. Hence, it is not so surprising that ADSC-based cell therapies are continuously being tested as an alternative, yet more effective, therapy in the management of musculoskeletal conditions.

Therefore, in this concise review, focusing on the type of MSDs subjected to therapeutic application of ADSCs in the recently published clinical studies, a brief characterization of MSDs as well as corresponding conventional therapeutic approaches including regenerative therapies using stem cell other than ADSCs will be discussed to provide an idea of under which circumstances and on what bases the ADSC-based cell therapy was implemented. By providing an overview of the current status of ADSC-based cell therapy on MSDs, we hope that this concise review can help to develop better and optimized strategies of ADSC-based therapeutics for MSDs as well as to find novel clinical applications of ADSCs in the near future.

## 2. MSD as a Major Target of ADSC-Based Cell Therapeutics

The PubMed search conducted on 1 June 2021 using “adipose derived stem cells or adipose derived regenerative cells or adipose derived stromal cells” as keywords with a filtering condition of article type “clinical trial” and a publication date of “10 years” came up with 167 studies. Among those articles, the number of original clinical studies that involved human subjects was 106, but 28 of them used stromal vascular fractions (SVFs) that are known to contain ADSCs [11] instead of isolated ADSCs. Since SVFs may be missing or without additional biological impact as compared to the standard cell therapy using isolated ADSCs only, those 28 studies are not covered by this review.

Out of a total of 78 studies, about one-third of the studies (21 studies) involved the application of ADSCs on MSCs, including but not limited to osteoarthritis, achilles tendinopathy, and rotator cuff tears. The top three disease categories also included gastrointestinal (14 studies) and circulatory (10 studies) categories (Figure 1). The details of each individual study covered by this review are described in the following subsections cartegorized by the type of musculoskeletal disorders (MSD) targeted.

## 3. Type of MSDs Targeted

### 3.1. Arthritis

Further analysis of the 21 clinical studies involving the use of isolated ADSCs on MSDs indicated that arthritis (10 osteoarthritis and 1 rheumatoid arthritis) was the most frequently targeted disorder of the musculoskeletal system. The clinical studies of ADSC-based cell therapy on arthritis are listed in Table 1. 

Classification by the anatomic sites affected (so targeted by the ADSC-based cell therapy) indicated that osteoarthritis (OA) of the knee was the major MSD with nine studies, and there were two studies for osteoarthritis of the ankle and refractory rheumatoid arthritis. As the most common musculoskeletal progressive condition, OA is a degenerative disease of the joints that displays clinical signs such as cartilage loss, osteophyte formation, and periarticular bone deformation [23]. Various pro-inflammatory cytokines and growth factors such as interleukin-1, tumor necrosis factor-alpha, transforming growth factor-beta, and matrix metalloproteinase are known to contribute to the progression of OA [24]. As conservative treatments, pharmacological agents such as acetaminophen, aspirin, and oral non-steroidal anti-inflammatory drugs (NSAIDs) are recommended for early management of OA [25] and surgical interventions such as total joint replacements may be necessary for severe OA with persisting pains [26]. However, pharmacological agents are not sufficient to correct the underlying structural abnormalities so that they are not able to prevent the progressive degeneration of the OA joint [27]. In the case of total joint replacement, although it is generally successful with enhanced mobility and reduced pain, it also has its own disadvantages such as a substantial risk of thrombosis and infection as a major surgical procedure and a high cost to cover hospital care and rehabilitation, which is similar to many other major surgeries [28]. Such limitations of conservative treatments promoted the development of less-invasive approaches for the management of OA.

Intra-articular injection of hyaluronic acid or platelet-rich plasma (PRP) [29,30], as well as MSCs, are the well-known examples of such less-invasive approaches. From the early 2000s, MSC-based cell therapy for OA has been suggested [31], and the emergence of MSC-based cell therapy in OA treatment is based on the ease of harvesting, the safety, and the cartilage differentiation potential of MSCs [13,32] and their paracrine and immunomodualtory effects [33,34,35]. The first clinical study that examined the effect of BM-MSC on articular cartilage defect was almost two decades old [36]. In that particular study, where the effect of BM-MSC transplantation was compared to that of a high tibial osteotomy in treating the articular cartilage defect, 42 weeks of transplantation resulted in hyaline cartilage-like tissue regeneration and improvement of both the arthroscopic and histological grading score, suggesting the clinical feasibility of MSC-based cell therapy for OA. Thereafter, such a beneficial effect of MSC-based cell therapy on OA has been further validated in many different pre-clinical and clinical studies [33], and it was adopted as a possible alternative to conventional therapeutics for treating other diseases as well [37].

Similarly, the effect of ADSCs on OA treatment has been investigated in animals of various species first, and after preclinical animal studies showed evidence of ADSC-mediated cartilage regeneration [38,39,40], the feasibility of using ADSCs for OA treatment in humans has been further scrutinized. One of the early studies tested the clinical potential of ADSCs in treating OA-utilized autologous ADSCS in the form of SVF with platelet-rich plasma and hyaluronic acid [41]. In this particular study, two human subjects with knee OA were treated, and regeneration of cartilage-like tissue was confirmed by magnetic resonance imaging (MRI). Thereafter, more studies using ADSCs for treating OA in humans, as indicated in Table 1, became available, accumulating evidence of cartilage regeneration.

The majority (eight out of nine) of the studies on OA utilized autologous ADSCs [12,13,14,15,17,18,19,20], and ADSCs were most frequently derived from abdominal fat (five out of nine) [13,15,18,19,20]. The number of ADSCs for a single injection ranged from 1.89 × 10^6^ to 1.0 × 10^8^, and the most frequently used dose was 5 × 10^7^ [15,17,20,22]. The method used to deliver ADSCs was intra-articular injection of ADSC solution, except the three studies either used platelet-rich plasma (PRP) [12] or fibrin glue [14] or partially demineralized cancellous bone [21] in combination with ADSC solution. No study on OA reported any treatment-related significant adverse events suggesting the safety of using ADSCs in treating OA. However, the study examined the effect of intravenously injected allogeneic ADSCs on refractory-rheumatoid-arthritis-reported adverse events (AEs)—although most of the AEs were of mild-to-moderate intensity [16]. In that particular study, transient fever was the most frequent treatment-related AE. Although it was not clearly determined, some form of infusion reaction was suggested as the underlying mechanism [42]. Additionally, there was one case of a lacunar infarction (left hemihypoesthesia and paretic ataxic gait), which was regarded as the dose-limiting toxicity. Since no apparent cause was determined, it was considered as likely treatment-related. Nevertheless, the treatment was well tolerated without dose-related toxicity, and it even demonstrated signs for potential therapeutic effects, calling for further research to investigate.

In terms of the clinical efficacy of ADSCs on OA, except studies with other primary purposes such as evaluating the safety of using ADSCs [15,16] or validating multi-compositional MRI as an effective tool for evaluating cartilage repair [22], most of the studies reported significantly improved pain and/or function. In the study where the patients received infrapatellar fat pad-derived MSCs with PRP, both the mean Lysholm score [43] and the Tegner activity scale [44], which measure activities of daily living, significantly increased in the experimental group compared to the control group that matched in terms of patient age and sex and follow-up period, suggesting improved knee function [12]. Furthermore, ADSC treatment significantly decreased the VAS score, indicating an improvement in the patient’s pain.

In another study, the safety and efficacy of autologous ADSCs without adjuvants indicated that treatment with 1.0 × 10^8^ ADSCs resulted in a 39% reduction of the WOMAC score, which measures pain, stiffness, and physical functioning of the joints [45] and a 45% decrease in the VAS at six months following injection. Furthermore, in radiological evaluation, it was found that the size of cartilage defects significantly decreased both in the medial femoral (40% decrease) and the tibial condyles (49% decrease) as well as in the lateral femoral (51% decrease) and the tibial condyles (46% decrease) at six months. Additionally, the cartilage volume significantly increased over the six months both in the medial femoral (14% increase) and the tibial condyles (22% increase) [13], suggesting regeneration of damaged cartilage. 

While these studies involved a single injection of ADSCs, there are also studies that utilized multiple injections of ADSCs [17,18]. For example, a study published in 2018 examined the long-term (96 weeks) safety and efficacy of repeated injection of ADSCs (with an interval of 48 weeks between the first two and the third injection), and the results indicated that the WOMAC score gradually reduced over time with a mean improvement rate of 27.81, 48.63, 39.07, 47.95, and 53.29%, at the 12th, 24th, 48th, 72nd, and 96th weeks following the initial injection, respectively. Furthermore, MRI evaluation showed that an increase in the cartilage thickness was more significant after the third injection compared with the first two injections, suggesting enhanced benefits of repeated injections [17]. Additional clinical-efficacy-related findings from other studies are also summarized in Table 1.

Altogether, these studies demonstrated that using ADSC-based cell therapy on OA is safe, and it produced promising results so that further clinical studies to verify its safety and efficacy as well as to set up a standardized therapeutic protocol are recommended. According to the ClinicalTrials.gov (accessed on 24 September 2021), 12 clinical trials to examine the effect of ADSCs on OA are ongoing (categories counted; recruiting, not yet recruiting, active, not recruiting, and enrolling by invitation) as of now.

In OA or degenerative join disease, damages to chondroblasts, chondrocytes, and the extracellular matrix (ECM) induced by various factors, such as oxidative stress, inflammatory factors, and mitochondrial dysfunction [46], initiate the degradation of cartilage tissue, which eventually leads to structural failure and loss of function [47]. Therefore, the fundamental premise of MSC-based therapeutic approaches for treating OA is that MSCs both/either directly adhere and become incorporated into the host tissue for osteogenic differentiation and/or exert reparatory effects on host cells via a paracrine mechanism, and empirical evidence indicated that those two mechanisms may synergistically work together [48].

For direct incorporation of MSCs in treating OA, two different studies have reported direct adherence and incorporation of injected MSCs, although they were not ADSCs. First, MSCs isolated from synovium of rats were used for meniscus cartilage regeneration, and the results indicated that the intra-articular injected MSCs migrated and adhered to the site of injury, and filled the meniscal defect [49]. In another study, umbilical-cord-blood-derived (UCB) MSCs with hyaluronic acid (HA) were utilized to treat rabbit joint articular cartilage defects, and the delivered UCB-MSCs adhered to the site of injury and regenerated cartilage comparable to normal cartilage tissue in terms of cellular structure and collagen organization [50]. Nevertheless, although those previous studies have clearly demonstrated that the delivered MSCs definitely attached at the site of injury, whether those incorporated MSCs were indeed differentiated into chondroblasts and/or chondrocytes is still inconclusive. In fact, other studies involved implanted a cell-tracking strategy that suggested that the adherence of MSCs at the site of cartilage defects was necessary, but those adhered MSCs were not necessarily differentiated into new chondroblasts and/or chondrocytes [48,51]. 

As for the paracrine effects, MSCs are known to secrete a wide range of bioactive factors, such as proteins, nucleic acids, proteasomes, exosomes, microRNA, and membrane vesicles, in response to the surrounding environment, and those bioactive factors affect various biological entities including the immune system, apoptosis, and growth and differentiation [52]. The secretome of MSCs can be categorized into the following three classes: growth factors, cytokines, and extracellular vesicles [53,54]. The growth factors and cytokines released from MSCs can be either pro-inflammatory or anti-inflammatory [54]. Vascular endothelial growth factor (VEGF), tumor necrosis factor β1 (TGF-β1), interleukin 13 (IL-13), and insulin-like growth factor (IGF-1) are some examples of anti-inflammatory mediators released from MSCs [55,56,57,58,59]. Although the secretome of MSC also includes pro-inflammatory mediators such as IL-1b, IL-6, IL-8, IL-9, and matrix metalloproteinase 3 (MMP-3) [60,61,62,63] and the final effect of MSCs on inflammation is decided by the temporal and spatial net effect of those growth factors and cytokines, mounting evidence indicates that MSCs more than often produces an overall anti-inflammatory effect [64,65,66]. 

It should be noted that although those growth factors and cytokines were categorized as pro- and anti-inflammatory mediators for the sake of discussion, the true nature of any given one of them is too complicated to be described by a single biological function. For example, IGF-I and TGF-β have been demonstrated to enhance the chondrogenic differentiation of MSCs [67], and they are also reported to increase the production of cartilage matrix components such as proteoglycan, type II collagen, and aggrecan in chondrocytes [68]. Furthermore, IGF-1 is known to regulate cellular apoptosis [69], and, in fact, it has been demonstrated that it can suppress apoptosis via the Src/PI-3K/AKT pathway in chondrocytes [70]. Therefore, it is most likely that those bioactive molecules from MSCs work together in a complex signaling network to produce an overall reparative impact on damaged cartilage. 

Speaking of the anti-apoptotic effect of MSC-derived bioactive molecules, it has been reported that co-culturing with MSCs decreased the expression of pro-apoptotic proteins such as caspase 3 and Bax, while it increased the expression of anti-apoptotic protein, Bcl-2, in alveolar macrophages [71], suggesting the anti-apoptotic effect of MSC-derived secretome. Additionally, evidence more directly related to the anti-apoptotic effect of ADSC-derived secretome on chondrocytes has been reported as well [72]. Based on those findings, it can be speculated that the growth factors and the cytokines released by ADSCs may prevent the death of chondrocytes, in addition to suppressing the inflammatory response in the diseased joint. Therefore, as the underlying mechanisms of ADSC-induced cartilage regeneration, it seems that the incorporation of stem both cells and paracrine effects contributes to the regenerative effect of MSCs, and it is likely that the same can be applied to ADSCs as well.

### 3.2. Craniomaxillofacial Defects

Out of 20 clinical studies involving the use of isolated ADSCs on musculoskeletal disorders, 3 studies were on craniofacial defects, namely, calvarial defects, caraniofacial microsomia, and cranio-maxillofacial hard-tissue defects (Table 2).

Although these three studies were classified in this category because they were dealing with craniofacial defects, they can be further classified by the characteristic of ADSCs with which those studies intended to produce therapeutic effects, including the osteogenic differentiation potential and the adipogenic differentiation potential. To be more specific, the osteogenic differentiation potential is for calvarial defects and cranio-maxillofacial hard-tissue defects, and the adipogenic differentiation potential is for craniofacial microsomia.

Calvarial defects refer to defects of the skull and are caused by various reasons including but not limited to trauma, infection, congenital malformations, neoplasm, and the surgical removal of tumors [76]. Likewise, cranio-maxillofacial hard-tissue defects can be a result of congenital malformations, traumatic avulsion, tumor resection, or severe infection. Clinical approaches for reconstruction of such defects encompass autografts, allografts, xenografts, or alloplastic grafts. Although the use of autografts is considered to be the gold standard for the reconstruction of bony defects [77,78], they still have limitations such as donor site morbidity, bone resorption, and lack of tissue availability [79,80,81]. On the other hand, alloplastic grafts have no donor-site morbidity and can be precisely shaped for individualized reconstruction so that they have replaced autografts as the more advanced gold standard for the reconstruction [82]. Nevertheless, they also have limitations such as a lack of ability to integrate and grow with the host bone [83,84]. 

To achieve proper regeneration, cell delivery to bony defect may be required in addition to the use of a proper scaffold. In general, cell-based therapeutic strategies for bony defects utilize either scaffolds pre-seeded with cells or acellular scaffolds promoting in situ recruitment of autologous cells [85]. Due to the limited quantity issue of BM-MSCs, ADSCs have become an alternative source of adult stem cells that are more easily obtainable in large numbers. Additionally, the osteogenic differentiation potential of ADSCs made them an even more promising candidate for bony defects.

The very first animal study on the feasibility of using ADSCs for calvarial defects was reported in 2004. In that particular study, it was demonstrated that ADSC seeded in a poly(lactic-co-glycolic acid) (PLGA) scaffold achieved complete bone bridging in a mouse model, and the treatment contributed to the majority (84–99%) of newly formed bone [86]. Furthermore, the first case report of the use of ADSCs to augment calvarial defects in a 7-year-old girl came in 2004 as well [87]. In that study, ADSCs with fibrin glue promoted new bone formation and near complete calvarial continuity within three months following the treatment. Slightly larger clinical studies followed some years later, and those studies are covered by this review.

In a study published in 2011, autologous ADSCs (1.5 × 10^7^ cells) derived from abdominal fat were used to augment calvarial defects in four patients [73]. For adjuvant, beta-tricalcium phosphate (betaTCP) granule, a well-known bone substitute material to improve osteogenesis [88,89], was used, and the results indicated satisfactory ossification without complications. In another study published in 2014, it was demonstrated that a composite of abdominal-fat-derived ADSCs seeded on resorbable scaffolds was applied to hard tissue defects of various craniomaxillofacial sites, including the frontal sinus (three cases), the frontal cranium (two cases), the parietal cranium (two cases), the temporal cranium (one case), the mandible chin (one case), the mandible body (two cases), and the nasal septum (two cases). The average number of ADSCs applied was approximately 6.5 × 10^6^ with a range of 2.8 × 10^6^ to 1.6 × 10^7^, and the results indicated that the composites successfully integrated into host tissues in 10 out of 13 cases [75].

Of note, unlike the majority of the OA cases where naive ADSCs were treated without any other adjuvants, those clinical trials for ossification frequently used scaffolds to promote new bone formation. This indicates that, although naive ADSCs have some beneficial effect to augment a bony defect [90], using only naive ADSCs for a bony defect may not be sufficient to produce the desired outcomes. Speaking of adjuvants for ossification, osteoinduction prior to transplantation is known to be effective, especially in treating critical-size defects [91,92]. Therefore, future studies on ADSCs, or any type of stem cells for that matter, for a bony defect may focus on finding an optimized combination of adjuvants such as scaffolds and osteoinducing agents. A brief overview on currently available options may help in selecting the ideal osteoinducing agents for future studies.

Although ADSCs have adipogenic, chondrogenic, or osteogenic differentiation potential, each specific lineage has major regulators, and Runx2 and Osterix are such major regulators for osteogenesis [93,94]. Moreover, several signaling pathways including bone morphogenetic protein (BMP) [95], Notch [96], Wnt [97], and Hedgehog-signaling [98] are known to regulate osteogenic differentiation, and the Wnt signaling pathway is possibly the most critical since it drives ADSCs away from adipogenic or chondrogenic lineages to a osteogenic lineage by increasing Runx2 and Osterix [99,100]. There are also well-known supplementary substances added to the medium to promote osteogenic differentiation of ADSCs such as dexamethasone, beta-glycerophosphate, and ascorbic acid [101]. Dexamethasone has been reported to increase Runx2 activity, Beta-glycerophosphate promotes osteogenesis by being a phosphate source, and ascorbic acid increases production and subsequent secretion of pro-collagen [102].

In addition, pro-osteogenic growth factors such as bone morphogenetic protein (BMP) can be used to promote osteogenesis. Belonging to the transforming growth factor (TGF) family [103], BMP is a well-established pro-osteogenic growth factor [104], and BMP-induced osteogenic differentiation of ADSCs has been reported [105,106]. The binding of BMP-2 or -3 to Ser/Thr kinase receptors results in phosphorylation of the Smad1/5/8 complex, which eventually increases the expression of RunX2 and Osterix by recruiting Smad4 to the complex [95]. The effectiveness of BMP in bone repair has been demonstrated in clinical trials [107,108]. Lastly, there are also other supplements that may enhance the osteogeneic potential of ADSCs, and they include, but are not limited to, vitamin D3 [109], selenium [110], and alendronate [111].

In addition to the above-mentioned biological molecules that regulate the osteogenic differentiation of ADSCs, using meticulously chosen biomaterials as scaffolds to provide mechanical stability and protection may also promote osteogenic differentiation of ADSCs. An appropriate scaffold can simulate the biological signals from extracellular the matrix, and it provides binding sites for cell adhesion as well as space for calcium deposition. To accommodate such requirements, carefully considering the size and interconnection of pores, as well as the stiffness of material (material property), are important. For example, although the size of osteoblasts is in the range of 10–50 μm [112], regeneration of mineralized bone was enhanced with macropores (100–200 μm) possibly because the lager pore size allows the infiltration of other cells involved in colonization and vascularization [113]. On the other hand, in contrast to macropores, micropores (pore size <10 μm) provide a greater surface area for cell adhesion and showed better bone protein adsorption [114].

Therefore, a wide range of materials with various pore sizes are actively being developed and tested [115,116]. The frequently reported biomaterials for the bone-regenerating scaffolds include hydroxyapatite (HA) [117], β-tricalcium phosphate (β-TCP) [73], synthetic polymers such as polylactic acid (PLA) [118], and poly lactic-co-glycolic acid (PLGA) [119]. Furthermore, a combination of osteogenic bioactive molecules and biomaterial-based scaffolds has also been tested. For example, a BMP-2 coated PLGA scaffold was applied to a calvarial defect, and the result indicated that the growth-factor-immobilized scaffold increased the osteogenic differentiation of ADSCs [120]. As such, stem cells, bioactive molecules, and scaffolds are three major components of the currently available bone regeneration strategy [121], and finding an ideal combination of these factors can make or break the current bone regeneration approaches. Since ADSCs harvested from patients with ongoing osteoporosis or aging may have a compromised potential for bone formation [122,123], these combinatory approaches will be especially beneficial when such patients undergo autologous ADSC therapy.

Craniofacial microsomia is one of the most common congenital conditions and is associated with anomalies of the jaws, ears, facial soft tissue, orbits, and facial nerve function [124]. Due to such a wide phenotypic spectrum, diagnosis and treatment of craniofacial microsomia is challenging. Often accompanied in craniofacial microsomia, soft tissue deficiency can be repaired with reconstructive techniques such as free flap, dermal fat graft, and structural fat graft. 

Also known as free tissue transfer, free flap surgery refers to a transplantation of tissue and its blood supply, which are surgically removed from one part of the body for the purpose of reconstruction. Although the adipofascial free flap is regarded as the best method to provide a large amount of soft tissue for a severe deficiency [125], limitations such as donor-site morbidity and a lengthy procedure still remain. Another well-established approach is a dermal fat graft for moderate and mild deformities. However, a certain degree of resorption and the subsequent need for additional augmentations are the limitations of this procedure [126]. Finally, there is the structural fat grafting that has changed the way various reconstruction-requiring conditions are treated [127,128]. Invented by Sidney Coleman [129], structural fat grafting involves fat harvesting from the abdomen, flanks, thighs, or buttocks; a refinement process; and microinjection (injection of small aliquots). The structural fat grafting can increase the precision of delivery, minimize scarring, and decrease donor-site morbidity, while yielding a greater number of viable adipocytes with more optimal function within fat grafts [130].

The study covered by this review also applied the structural fat grafting on microsomia with modification. The modification was to enrich the fat grafts with additionally isolated ADSCs so that the ADSC content of the graft increased [74]. In fact, ADSC-enriched fat transplantation significantly increased the surviving fat volume in that study, and such results stand to reason considering ADSCs are more tenacious than mature adipocytes, and the surviving ADSCs are the major effector of fat tissue transplantation [131,132].

Considering ADSCs can also differentiate into endothelial cells and smooth muscle cells, leading to new blood vessel formation [133,134], such a beneficial effect of ADSCs might be the result of increased neovascularization within the transplanted fat tissue [135]. Additionally, ADSCs are known to secrete pro-angiogenic factors such as VEGF and hepatocyte growth factor (HGF) [136], and therefore, ADSCs may promote angiogenesis of the host tissue and within the graft to improve graft revascularization and subsequent graft survival. Another soluble factor released from ADSCs that may improve fat graft survival is IGF-1. IGF-1 is a known anti-apoptotic factor [137], and it has been demonstrated that sustained release of IGF-1 from ADSCs protected cardiomyocytes from apoptosis following myocardial infarction [138].

Furthermore, IGF-1 is also a potent mitogen for adipocyte differentiation [139], and therefore, ADSCs may prevent apoptosis of the graft, while stimulating preadipocytes’ differentiation into mature adipocytes, thus retaining graft volume. Lastly, the most recent mechanistic insight for the role of ADSCs on the survival of fat graft involves the extracellular vesicles (EVs) released from ADSCs [140]. In that particular study, it was demonstrated that ADSC-derived EVs enriched with the let-7 family of miRNAs improved the survival of fat grafts by promoting angiogenesis via the let-7/argonaute 1 (AGO1)/VEGF signaling pathway.

### 3.3. Tendon- and Ligament-Related Disorders

Out of 20 clinical studies covered by this review, 5 studies were on tendon- and ligament-related disorders, namely, 2 studies on lateral epicondylosis and lateral elbow tendinopathy, 2 studies on rotator cuff tears, and 1 study on anterior cruciate ligament (Table 3).

Often called the “tennis elbow,” lateral epicondylosis is a common painful condition that affects the tendons joining the forearm muscles on the outside of the elbow [146]. Common non-surgical treatment of lateral epicondylosis includes, but not is limited to, physiotherapy [147], injections of corticosteroid [148], extracorporeal shock wave therapy [149], acupuncture [150], topical glyceryl trinitrate [151], botulinum toxin [152], and platelet-rich plasma [153]. Regarding the cell therapy for lateral epicondylosis, it was surprising that only a handful of studies had been conducted to date. Except the two studies subjected to this review, there were only five clinical studies that involved cell therapy on lateral epicondylosis, and they have used tenocytes, tenocyte-like cells, bone marrow aspirates, and allogeneic ADSCs, respectively [154,155,156,157,158]. It was also interesting to notice that tenocytes or tenocyte-like cells were more frequently used than MSCs [155,157]. 

Considering lateral epicondylosis involves inflammation and/or tearing of the tendons joining the forearm muscles on the outside of the elbow [76], it seems to be intuitively logical to prefer tenocyte or tenocyte-like cells over classical stem cells such as BM-MSCs. Since the one study used bone marrow aspirate rather than isolated BM-MSCs [158], the studies covered by this review are the only studies that examined the effect of isolated stem cells. In the study published in 2015, either 10^6^ or 10^7^ allogeneic ADSCs were used, and the results showed that there was no significant AEs and VAS score that progressively decreased, indicating improved pain management. Furthermore, the MEPI performance score significantly increased, demonstrating that ADSC therapy was safe and effective for treating lateral epicondylosis [141].

Another study where 18 tennis players were treated with approximately 8 × 10^6^ autologous ADSCs for recalcitrant lateral elbow tendinopathy indicated that ADSC therapy significantly improved clinical parameters readily at one month following injection and structural repair of the origin of common extensor tendon at six months after injection [142], without any joint effusion or skin hypersensitivity reactions that had been observed in the previous similar study where allogeneic ADSCs were used [141].

Another tendon-related disorder treated with ADSCs is rotator cuff tears. The rotator cuff refers to a structured tendinous insertions of muscles for stabilizing the glenohumeral joint, and rotator cuff disorder is the most common condition of the shoulder in the aged population [159,160]. Rotator cuff tears can be caused by either traumatic or degenerative reasons. Traumatic tears occur literally due to significant trauma to the rotator cuff, while degenerative tears are more common and multifactorial in that both extrinsic factors such as the anterior part of the cuff abutting against the coracoacromial arch during forward elevation of the shoulder [161] and intrinsic factors like excess levels of reactive oxygen species (ROS) damaging tendons [162] gradually lead to a full-thickness tear.

Rotator cuff tears are often treated with surgical options to increase function and decrease pain, and new materials and surgical techniques to improve the outcomes of the surgical repair have been utilized to meet such ends [163]. However, the regeneration of the fibro-cartilaginous transition zone between the rotator cuff and the bone has not been satisfactory [164] with a persistently high failure rate of the repair [165]. Thus, the efforts to improve the biological environment around the damaged cuff using growth factors as well as stem cells came into play [166,167]. 

Regarding the use of stem cells for rotator cuff tears, it was demonstrated that rotator-cuff-derived MSCs have higher myogenic potential compared to BM-MSCs [168], and tenocyte-derived stem cells from tendons have been isolated and characterized [169]. However, the very first clinical study of stem cells’ effect on rotator cuff utilized mononuclear stem cells from bone marrow aspirate, which resulted in better functional outcomes than would usually be expected without stem cell adjuvant [170]. Similarly, the study used additional autologous ADSCs injection (a mean of 4.46 × 10^6^ cells in 2 mL of fibrin glue) during arthroscopic rotator cuff repair, which also demonstrated that using ADSCs as the adjuvant improved the structural outcome possibly by providing an adequate biological environment around the cuff [143]. Furthermore, not only as an adjuvant but also as an independent therapeutic, the effect of ADSCs in the treatment of rotator cuffs has been demonstrated in a very recent study [144]. In that study, a single injection of an average of 11.4 × 10^6^ autologous ADSCs was used to treat symptomatic, partial-thickness rotator cuff tears, and the treatment improved shoulder function without AEs suggesting a potential of ADSCs as a substitute for corticosteroids commonly used for short-term pain relief [171]. Altogether, those studies indicated that ADSC-based cell therapy represents a feasible option in order to improve rotator cuff regeneration. 

As to the possible underlying mechanisms of MSCs for treating tendinopathy, although initial therapeutic strategy might have been targeted tenogenic differentiation of delivered MSCs, it seems that the therapeutic effects of MSCs are mainly achieved by interacting with the tendon resident cells [172]. It has been reported that co-culturing tendon cells and BM-MSCs up-regulated the expression of tendon-related genes such as scleraxis and tenomodulin, collagens, decorine, and tenascin, leading to significant tendon ECM deposition [173,174], which suggests that BM-MSC may enhance the tenogenic capacities of tendon-derived stem cells and tendon stem/progenitor cells. Similar to the case of BM-MSCs, ADSCs also have demonstrated that they communicate with tendon cells to increase the expression of tendon-related genes [175,176]. Furthermore, a previous study that examined the effect of ADSCs on the tendon niche showed that ADSCs helped to preserve the native architecture of tendon tissue with early increased collagenolytic activity of matrix metalloproteinases (MMPs) [177]. Considering the deposition of ECM increased in ADSC and a human tendon-derived cell co-culture system [176], it may be possible that ADSCs exert a beneficial effect on tendon regeneration by modulating the microenvironment of the tendon niche.

Finally, an anterior cruciate ligament (ACL) tear is the disorder that had been subjected to ADSC-based cell therapy. With the posterior cruciate ligament, the anterior cruciate ligament helps stabilize a knee joint [178]. An anterior cruciate ligament tear is one of the most common knee injuries, and it is estimated that approximately 80,000 to 100,000 anterior cruciate ligament repairs are performed each year in the United States [179]. The majority of anterior cruciate ligament tears are caused by a non-contact mechanism such as sudden directional change making the knee rotate inward [180]. Due to greatly advanced surgical techniques [181], anterior cruciate ligament reconstruction became a gold standard for anterior cruciate ligament tears [182]. However, only less than a half of patients achieve full recovery [183], and patellar tendon grafts used for reconstruction are different from natural anterior cruciate ligaments [184]. Such limitations of anterior cruciate ligament reconstruction necessitated exploration of other therapeutic options including the use of growth factors, platelet-rich plasma, and stem cells. 

A number of different types of stem cells have been examined for their potential as alternative therapeutics for anterior cruciate ligament tears to date. First of all, MSCs have been most widely investigated possibly because they are capable of ligamentogenic differentiation with proper growth factors [185,186,187]. Another type of MSC examined for their potential in treating anterior cruciate ligament tears is synovium-derived mesenchymal stem cells (SMSCs) [188]. Furthermore, it has been demonstrated that they have higher proliferation and differentiation potentials than MSCs derived from other tissues [189], suggesting SMSCs can be a feasible candidate for alternative therapeutics. Not surprisingly, stem cells derived from anterior cruciate ligaments were examined for their potential as well, and the studies indicated that anterior cruciate ligament-derived stem cells have characteristics very similar to those of BM-MSC, suggesting they could be a viable source of stem cells for anterior cruciate ligament repair [190,191]. 

Regarding the potential use of ADSCS for the treatment of anterior cruciate ligament tears, in vitro studies demonstrated that human ADSCs failed to stimulate anterior cruciate ligament fibroblast proliferation and collagen production [192], which had been observed for porcine ADSCs [193], making the therapeutic beneficial of ADSCs in treating anterior cruciate ligament tears uncertain. In fact, one study covered by this review also demonstrated that 1.8 × 10^7^ ADSCs loaded in bone-patellar tendon-bone (BTB) graft produced no statistically significant improvement compared to a control group (without ADSC administration). Nevertheless, there are only a handful of studies that exist, so further studies are needed to fully evaluate the potential of ADSCs as alternative therapeutics for anterior cruciate ligament tears.

Although how ADSCs enhance ligament repair is not completely elucidated, mechanical signals from the host tissue may significantly contribute to the observed beneficial effect of ADSCs. A previous study examined the effect of co-culturing with ACL cells, and mechanical stress on MSC indicated that the combination of regulatory signals from ACL cells and mechanical stress enhanced selective differentiation of MSCs toward ligament cells by demonstrating increased typical ACL cell markers such as collagen type I and III and tenascin in MSCs following co-culturing with ACL cells in the presence of mechanical stress [194]. Therefore, it is reasonable to speculate that mechanical signals from host tissue following cell transplantation may stimulate cell-surface stretch receptors and adhesion sites, resulting in increased synthesis and secretion of key ligament ECM components [195].

### 3.4. Spine Disorders

Out of 20 clinical studies covered by this review, 2 studies were on spine disorders, namely, one on degenerative spondylolisthesis and the other on chronic discongenic low back pain (Table 4).

Spinal disorder refers to a condition impairing the backbone, and the associated pain poses a major medical and socioeconomic problem due to its high prevalence in the general population [198]. Degenerative spondylolisthesis is defined as a condition where one vertebral body slips forward on top of another one below without rupture of the posterior arc [199], and more than 10% of the population in the United States suffers from this condition [200]. For patients who are symptomatic, non-operative conservative treatment options includes anti-inflammatory medications such as non-steroidal anti-inflammatory drugs (NSAIDs) and narcotic analgesics [201], physical therapy [202], and epidural steroid injections [203]. On the other hand, fusion surgery has been proven to be effective for patients who do not respond well to non-operative management [204]. However, fusion surgery that frequently involves the use of autologous iliac crest bone grafts (AICBG) [205] accompanies a risk of complications such as donor site pain, infection, hematoma, and meralgia paresthetica [206,207]. 

In an effort to overcome such limitations, alternative options such as new bone substitutes [208] and stem-cell-based approaches have been explored. Studies that explored the potential of MSCs for spinal fusion have demonstrated that the use of MSCs produced outcomes comparable to those of iliac crest grafts in terms of histology and mechanical properties [209,210]. Regarding clinical trials involving application of ADSCs to degenerative spondylolisthesis, one study covered by this review reported that the use of 3D graft made of autologous ADSCs in patients receiving minimally invasive transforaminal lumbar interbody fusion (MI-TLIF) resulted in significant improvement in the VAS score and the ODI and achievement of grade 3 fusion without donor site complications [196]. Although only one study did not guarantee the clinical benefit of using ADSCs for degenerative spondylolisthesis so that further studies are required, a very recent study where a combination of BM-MSC and allogeneic graft achieved a higher rate of posterior spinal fusion and radiographic complete response without significant AEs [211] also suggests the potential of ADSCs, or stem cells in general, as alternative therapeutics for degenerative spondylolisthesis. 

Chronic low back pain is one of the leading causes of disability and one of the major clinical and socioeconomic global health burdens [212]. The term discogenic back pain refers to back pain caused by internal structural change of the lumbar intervertebral disc (IVD) without herniation, anatomical deformity, or other alternate clear causes of pain and disability [213]. Quite similar to the case of degenerative spondylolisthesis, common non-operative modalities for chronic discongenic low back pain include drug therapy using NSAIDs [214] and opioids [215], physical rehabilitation theary [216], epidural injection [217], and percutaneous intradiscal therapies to alter the internal mechanics of the disc with heat, radiofrequency, or injection of various chemicals [218]. As for surgical treatment options, interbody fusion [219], prosthesis replacement [220], and dynamic fixation system [221] have been utilized for chronic discogenic low back pain.

Regenerative approaches for chronic low back pain involve the use of PRP [222,223], chondrocytes [224], and stem cells. The feasibility and safety of MSC-based cell therapy for chronic back pain has also been examined, and clinical benefits such as analgesic effects and functional improvement were demonstrated [197,225,226]. Those studies also include the study using ADSCs, and according to that particular study, a single intradiscal injection of hyaluronic acid (HA) derivatives and an autologous ADSC (2 or 4 × 10^7^ cell/disc) mixture significantly improved VAS, ODI, and SF-36 scores without AEs in patients with chronic discogenic low back pain [197], demonstrating the safety and tolerability of ADSC-based cell therapy as well as its promising clinical efficacy.

The IVD is located between the vertebral bodies of the spinal column, and the nucleus pulposus (NP, nucleus) comprises the inner gelatinous structure [227]. The deterioration of NP architecture, which can be characterized by the change of gelatinous, hydrated ECM into a more fibrous tissue due to aging or pathologic trauma, is the main cause of intervertebral disc degeneration (IDD) [228], and IDD is a major cause of back pain. Therefore, restoring the adequate NP architecture may relieve the pain, and this is what MSC-based cell therapy targets. In other words, MSCs may differentiate into mature cells, support resident cell activity by paracrine mechanism, and/or recruit local progenitor cells to promote endogenous repair of the degenerated IVD [229]. As supporting evidence for such speculation, first, multiple studies have reported NP-cell-like differentiation of MSCs [230,231,232], and one recent study indicated that the IVD-transplanted autologous BM-MSCs survived in the host IVD tissues up to eight months [233]. 

Regarding the MSC supporting of resident cells, it has been demonstrated that co-culturing MSCs with NP cells increased cell proliferation and ECM production of NP cells [234,235], and the exosome-mediated paracrine mechanism was found to contribute to this resident-cell-supporting effect of MSCs [236]. Additionally, immunomodulation by MSCs may also contribute to the regenerative effect of MSCs in IDD. As for the inflammatory response during IDD, IVD resident cells and immune cells release several proinflammatory cytokines including, but not limited to, IL-1b, tumor necrosis factor alpha (TNFα), interferon-c, and prostaglandin E2 causing ECM breakdown, neoangiogenesis, and the stimulaton of additional cytokines [237,238,239]. Since such inflammatory stimuli can increase cell apoptosis and neurogenic differentiation of NP-MSCs, which may contribute to IVD reinnervation and the development of chronic LBP [240], MSCs that are known to produce anti-inflammatory cytokines, anticatabolic mediators, and growth factors even under IDD-like conditions [241,242] may help to create a microenvironment favoring IVD regeneration.

## 4. Conclusions

MSDs are now the second most common cause of years lived with disability (YLD) in the world [243]. Therefore, an effective long-term solution for MSDs would relieve pain and agony for a significant portion of the population as well as reduce the related socioeconomic costs for the healthcare system. Regenerative therapies, including stem cell-based therapy, are non-surgical conservative interventions that will likely be preferred over conventional invasive surgical approaches once their safety, reliability, and efficacy are confirmed in humans. The use of ADSC-based therapy in the regeneration of musculoskeletal tissue is relatively young but dynamic. Although the current literature regarding the clinical use of ADSCs in MSDs is still limited, the preliminary results of initial clinical trials both in humans and in animals indicate that it can be safe and effective for bone defects, cartilage regeneration, and tendinopathies (Figure 2). Nevertheless, to harness the full therapeutic potential of ADSCs, further basic studies are desired, as are longer-term safety studies and more randomized larger-scale controlled trials to examine the safety and efficacy of ADSC-based therapy for MSDs.

## Figures and Tables

**Figure 1 ijms-22-10586-f001:**
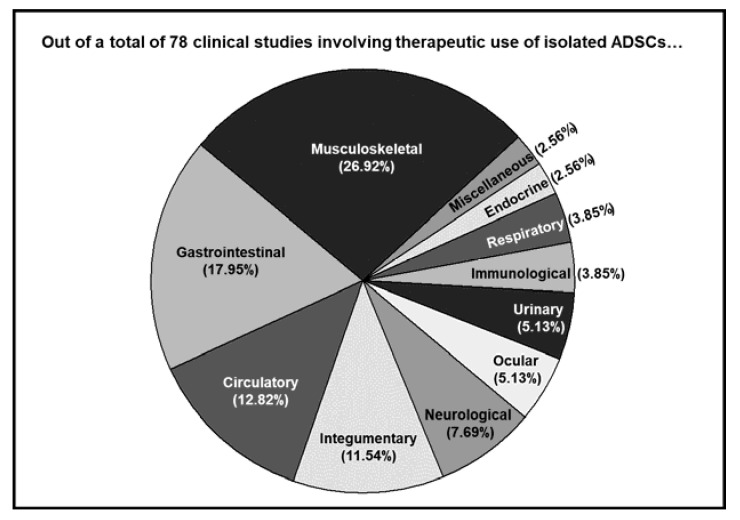
Different disorders subjected to isolated ADSC-based therapeutics are categorized by the systems the targeted tissue/organ belong to, and the ratio of each category is indicated.

**Figure 2 ijms-22-10586-f002:**
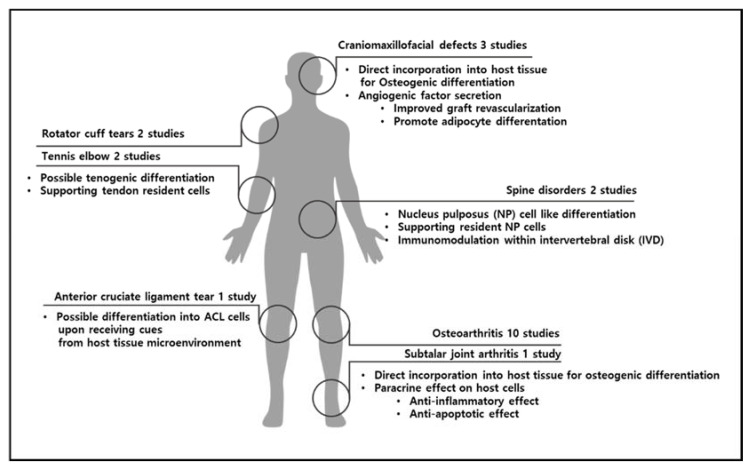
Summary of MSDs covered by this review and possible underlying mechanisms with which delivered ADSCs produce regenerative effects.

**Table 1 ijms-22-10586-t001:** Summary of clinical studies examined therapeutic use of hADSCs on arthritis for the last 10 years.

	Target Disorder	No. of Patients Treated (Age: Mean ± SD)	ADSC Type	ADSC Delivery	Study Outcome	Year	Ref. No.
1	Knee osteoarthritis	25 (54.20 ± 9.30)	Autologous, derived from infrapatellar fat pad	Infrapatellar injection with PRP ^1^	No major AEs ^2^ with significantly increase Lysholm, Tegner activity scale, and VAS ^3^ score	2012	[12]
2	Knee osteoarthritis	18 (63.0 ± 12.49)	Autologous, derived from abdominal fat	Intra-articular injection	Improved function and pain without adverse events, reduced cartilage defects	2014	[13]
3	Knee cartilage defects	40 (38.75 ± 9.56)	Autologous, derived from buttocks	After microfracture, intra-articular injection with fibrin glue	Improved KOOS ^4^ pain and symptom sub score	2016	[14]
4	Knee osteoarthritis	18 (64.63 ± 9.37)	Autologous, derived from abdominal fat	Intra-articular injection	Safe, without SAEs ^5^	2016	[15]
5	Refractory rheumatoid arthritis	46 (53.96 ± 20.64)	Allogeneic (Cx611)	Intravenous injection	Well tolerated with no evidence of dose-related toxicity, but some AEs and SAEs	2017	[16]
6	Osteoarthritis	18 (54.8 ± 17.73)	Autologous, isolated from lipoaspirates	Intra-articular injection, repeated	Safe and improved pain, function, and cartilage volume	2018	[17]
7	Knee osteoarthritis	20 (54.65 ± 11.99)	Autologous, derived from abdominal fat	Intra-articular injection	No SAEs, clinically significant pain, and functional improvement	2019	[18]
8	Knee osteoarthritis	12 (62.25 ± 6.50)	Autologous, derived from abdominal fat	Intra-articular injection	Significant improvement of the WOMAC ^6^ score without SAEs	2019	[19]
9	Knee osteoarthritis	26 (55.03 ± 9.19)	Autologous, derived from abdominal fat	Intra-articular injection	Significant improvements in joint function, pain, quality of life, and cartilage regeneration	2019	[20]
10	Subtalar joint arthritis	52 (56.9: 20.3–79.6) ^6^	Allogeneic	ADSC loaded, partially demineralized cancellous bone (AlloSource) was grafted	Good clinical outcomes in spite of the high non-union rates	2019	[21]
11	Knee osteoarthritis	18 (54.77 ± 17.79)	Allogeneic	Intra-articular injection	a possible compositional changes of cartilage, significant reduction in WOMAC ^7^ and SF-36 ^8^ scores	2019	[22]

^1^ Platelet-rich plasma (PRP), ^2^ adverse events (AEs), ^3^ visual analog scale (VAS), ^4^ knee injury and osteoarthritis outcome score (KOOS), ^5^ serious adverse events (SAEs). ^6^ In this study, age was presented as means with the range. ^7^ Western Ontario and McMaster Universities Osteoarthritis index (WOMAC), ^8^ Medical Outcomes Short-Form-36 questionnaire (SF-36).

**Table 2 ijms-22-10586-t002:** Summary of clinical studies involving the use of isolated hADSCs on craniomaxillofacial defects for the last 10 years.

	Target Disorder	No. of Patients Treated (Age: Mean ± SD)	ADSC Type	ADSC Delivery	Study Outcome	Year	Ref. No.
1	Calvarial defects	4 (63.8: 59–75) ^1^	Autologous, derived from abdominal fat	ADSCs containing betaTCP ^2^ granules were laid on the dura	No complications with satisfactory ossification	2011	[73]
2	Craniofacial microsomia	7 (12.10 ± 2.20)	Autologous, derived from abdominal fat	Subcutaneous injection in a form of ADSC-enriched fat	Significant increase of surviving fat volume	2013	[74]
3	Cranio-Maxillofacial hard-tissue defects	13 (53.23 ± 10.29)	Autologous, derived from abdominal fat	ADSC seeded resorbable scaffolds were implanted	Successful integration of the construct in 10 of the 13 cases.	2014	[75]

^1^ In this study, age was presented as mean with the range, ^2^ beta-tricalcium phosphate (betaTCP).

**Table 3 ijms-22-10586-t003:** Summary of clinical studies involving the use of isolated hADSCs on arm disorders for the last 10 years.

	Target Disorder	No. of Patients Treated (Age: Mean ± SD)	ADSC Type	ADSC Delivery	Study Outcome	Year	Ref. No.
1	Lateral epicondylosis	12 (51.85 ± 13.86)	Allogeneic	Intratendinous injection with fibrin glue	Safe and improved elbow pain (VAS ^1^), performance (MEPI ^2^), and structural defects	2015	[141]
2	chronic lateral elbow tendinopathy	18 (46.5 ± NA ^3^)	Autologous, derived from periumbilical zone	Percutaneous injection to the affected elbow	Significantly improved mean VAS scores for maximum pain score, QuickDASH ^4^-Compulsory score, QuickDASH-Sport score	2021	[142]
3	Rotator cuff tears	72 (59.05 ± 3.60)	Autologous, derived from buttocks	ADSC loaded in fibrin glue was injected	Although ADSC significantly improved structural outcomes in terms of the retear rate, there were no clinical differences compared to the control group	2017	[143]
4	Rotator cuff tears	11 (64.60 ± 9.60)	Autologous, derived from either the periumbilical abdominal area, bilateral flanks, or medial thigh fat	Intra-articular injection	Significantly higher mean ASES ^5^ total scores without adverse events	2020	[144]
5	Anterior cruciate ligament reconstruction	20 (24.70 ± 4.70)	Autologous, derived from abdominal or inner thigh fat	Intra-articular injection, applied to BTB ^6^ autograft	Although ADSC significantly improved knee function and healing/maturation of the graft, it was not significantly different compared to the control group	2019	[145]

^1^ Visual analog scale (VAS); ^2^ Mayo clinic performance index (MEPI); ^3^ not available; ^4^ mean quick disabilities of the arm, shoulder, and hand; ^5^ American Shoulder and Elbow Surgeons Standardized Shoulder Assessment Form (ASES); ^6^ bone-patellar tendon-bone (BTB).

**Table 4 ijms-22-10586-t004:** Summary of clinical studies involving the use of isolated hADSCs on spine disorders for the last 10 years.

	Target Disorder	No. of Patients Treated (Age: Mean ± SD)	ADSC Type	ADSC Delivery	Study Outcome	Year	Ref. No.
1	Degenerative spondylolisthesis (TLIF) ^1^	3 (48.70 ± 14.30)	Autologous, derived from abdominal fat	ADSC seeded DBM ^2^ was implanted into the disc space	Grad 3 fusion, VAS and ODI ^3^ improved	2017	[196]
2	Chronic discogenic low back pain	10 (43.50 ± 10.16)	Autologous, derived from abdominal fat	Percutaneous injection of ADSC in combination with HA ^4^ derivatives	Safe and tolerable without no adverse events.	2017	[197]

^1^ Transforaminal lumbar interbody fusion (TLIF), ^2^ demineralized bone matrix (DBM), ^3^ Oswestry disability index (ODI), ^4^ hyaruronic acid (HA).

## Data Availability

Not applicable.

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
