# Peer review of "ADSC-Based Cell Therapies for Musculoskeletal Disorders: A Review of Recent Clinical Trials"

_ijms, 2021, doi:10.3390/ijms221910586_

Round 1

Reviewer 1 Report

Dear authors,

thank you for your article, which I found quite informative. However, the article was very generally written and you only scratched the surface. I would recommend to outline and discuss the therapeutic effect of the different types of ADSCs stronger. Moreover, I would recommend to explain the effect of ADSCs on a more molecular level since this is the journal of molecule science.

However, I feel that the article in its current form does not have that much quality to be published in a journal with IF 5.9 since it is only a concise review and not a systematic review. Without going too much into detail there are some parts which you need to correct:

l.11 Please define ADSCs before first use.

  1. 17- l.23 These lines are very generally written and could be part of any abstract. You need to emphasize some aspects of review.

l.40 /l.48 repetition. The sentence in line 48 does not have any further value.

l.79. PLease give the exact time when the literature research has been performed.

l.167 this is too general. Please try to point out some exact improvements ( improved waling distance etc..)

l.425 please give reference

l.435 „and so forth“ does not sound very scientific

Author Response

Reviewer1

Dear authors,

Thank you for your article, which I found quite informative. However, the article was very generally written and you only scratched the surface. I would recommend to outline and discuss the therapeutic effect of the different types of ADSCs stronger. Moreover, I would recommend to explain the effect of ADSCs on a more molecular level since this is the journal of molecule science.

Response: The reviewer recommended to add 1) discussion on the therapeutic effect of the different types of ADSCs and 2) explanation on the effect of ADSCs at molecular level. It is clear that those subject are worthwhile to explore. Nevertheless, we could not fully accommodate the reviewer’s recommendation and only comply with the latter request for the following reason.

Comparing therapeutic effects of the different types of ADSCs (i.e., autologous vs. allogeneic) was not considered when selecting clinical studies for this particular review. Furthermore, no study covered by this review examined therapeutic effects of the different types of ADSCs in a single study. Therefore, it is almost impossible to compare and discuss the therapeutic effects of the different types of ADSCs based on studies independently conducted with either only autologous ADSCs or only allogeneic ADSCs.

However, we did explore possible molecular mechanisms of the reported beneficial effects of ADSC treatment as the reviewer recommended, and they are now discussed in each section where it is appropriate.

l.11 Please define ADSCs before first use.

Response: ADSCS is defined in the abstract as the reviewer recommended.

17- l.23 These lines are very generally written and could be part of any abstract. You need to emphasize some aspects of review.

Response: The line 17-23 has been revised to describe the specifics of this review, including possible mechanisms, as the reviewer recommended

l.40 /l.48 repetition. The sentence in line 48 does not have any further value.

Response: Remarks related to the use of less invasive procedure for obtaining ADSCs from the line 40 has been revised to avoid unnecessary repetition.

l.79. PLease give the exact time when the literature research has been performed.

Response: The literature search as conducted on June 1st, 2021, and the date of literature search is indicated in the revised manuscript as the reviewer requested.

l.167 this is too general. Please try to point out some exact improvements ( improved waling distance etc..)

Response: Some of the detailed information on the nature of improvement has been added as the reviewer recommended.

l.425 please give reference

Response: corresponding reference has been added. The world-wide burden of musculoskeletal diseases: a systematic analysis of the World Health Organization Burden of Diseases Database (http://dx.doi.org/10.1136/annrheumdis-2019-215142)

l.435 „and so forth“ does not sound very scientific

Response: the expression “and so forth” has been removed as the reviewer suggested.

Thanks to the reviewers’ valuable comments, our manuscript has been much improved and embellished. We authors deeply appreciate the reviewer’s valuable time and effort to review our manuscript. Revised parts of the manuscript are indicated in RED

Reviewer 2 Report

The authors presented a concise revise on the topic of ADSC-based cell therapies for musculoskeletal disorders. This review paper gives a very clear picture of using adipose-derived stem cells in different clinical applications. Overall, I think this paper should be interested in the researchers in this field. But, some questions and errors are still needed to be answered and revised.

(1) The full names of abbreviations should appear for the first time shown in the manuscript, such as ADSCs in the abstract, PRP (line 126), etc. Please check throughout the manuscript and correct it.

(2) Line 41, “…such as liposuction and liposuction”. Is it correct?

(3) As point 1, the full name of AEs must show in line 159.

(4) Line 175, a dot is missing at the end of this sentence.

(5) Line 415, the full name of PRP is not necessary here.

(6) This review focuses on the recently published clinical trials involving the use of ADSCs on musculoskeletal disorders. From these tables, these studies report different numbers of patients. Actually, the characteristics of patients may be also important, especially the age, to the effect of therapy. I recommend the authors provide information about the age range of patients in different reports.

Author Response

Reviewer 2

The authors presented a concise revise on the topic of ADSC-based cell therapies for musculoskeletal disorders. This review paper gives a very clear picture of using adipose-derived stem cells in different clinical applications. Overall, I think this paper should be interested in the researchers in this field. But, some questions and errors are still needed to be answered and revised.

(1) The full names of abbreviations should appear for the first time shown in the manuscript, such as ADSCs in the abstract, PRP (line 126), etc. Please check throughout the manuscript and correct it.

Response: Both ADSCs and PRP have been indicated with full names as the reviewer pointed out.

(2) Line 41, “…such as liposuction and liposuction”. Is it correct?

Response: First of all, it was a typo. Since other reviewer pointed out that the remarks regarding the use of less invasive procedure for obtaining ADSCs were repeated in line 40 and in line 48, the line 40 has been revised to avoid unnecessary repetition.

(3) As point 1, the full name of AEs must show in line 159.

Response: The full name of AEs has been spelled out as the reviewer requested.

(4) Line 175, a dot is missing at the end of this sentence.

Response: A missing dot has been added.

(5) Line 415, the full name of PRP is not necessary here.

Response: The full name of PRP has been removed.

(6) This review focuses on the recently published clinical trials involving the use of ADSCs on musculoskeletal disorders. From these tables, these studies report different numbers of patients. Actually, the characteristics of patients may be also important, especially the age, to the effect of therapy. I recommend the authors provide information about the age range of patients in different reports.

Response: The age range of patients has been added to the corresponding tables showing characteristics of patients

Thanks to the reviewers’ valuable comments, our manuscript has been much improved and embellished. We authors deeply appreciate the reviewer’s valuable time and effort to review our manuscript. Revised parts of the manuscript are indicated in RED

Reviewer 3 Report

I thank the authors for their hard work on this piece of research.

You nicely summarise the field and it will be useful for others moving forward.

My only comment would be to address the colours on the pie chart to allow for colour blind readers to follow. Additionally more figures would highlight the written work

Author Response

Reviewer 3

I thank the authors for their hard work on this piece of research.

You nicely summarize the field and it will be useful for others moving forward.

My only comment would be to address the colors on the pie chart to allow for color blind readers to follow. Additionally more figures would highlight the written work

Response: The color of the pie chart has been changed to graded black/white figure for the convenience of color-blinded readers, and an additional figure summarizes the application of ADSCs and possible underlying mechanisms has been added as the reviewer recommended.

Thanks to the reviewers’ valuable comments, our manuscript has been much improved and embellished. We authors deeply appreciate the reviewer’s valuable time and effort to review our manuscript. Revised parts of the manuscript are indicated in RED